# Delays, Detours, and Forks in the Road: Latent State Models of Training Dynamics

**Michael Y. Hu**                                          *michael.hu@nyu.edu*
*New York University*

**Angelica Chen**                                          *ac5968@nyu.edu*
*New York University*

**Naomi Saphra**                                          *ns4008@nyu.edu*
*New York University*

**Kyunghyun Cho**                                          *kyunghyun.cho@nyu.edu*
*New York University*
*Prescient Design, Genentech*
*CIFAR LMB*

**Reviewed on OpenReview:** *https://openreview.net/forum?id=NE2xXWo0LF*

## Abstract

The impact of randomness on model training is poorly understood. How do differences in data order and initialization actually manifest in the model, such that some training runs outperform others or converge faster? Furthermore, how can we interpret the resulting training dynamics and the phase transitions that characterize different trajectories? To understand the effect of randomness on the dynamics and outcomes of neural network training, we train models multiple times with different random seeds and compute a variety of metrics throughout training, such as the $L_2$ norm, mean, and variance of the neural network's weights. We then fit a hidden Markov model (HMM; Baum & Petrie, 1966) over the resulting sequences of metrics. The HMM represents training as a stochastic process of transitions between latent states, providing an intuitive overview of significant changes during training. Using our method, we produce a low-dimensional, discrete representation of training dynamics on grokking tasks, image classification, and masked language modeling. We use the HMM representation to study phase transitions and identify latent "detour" states that slow down convergence.

## 1 Introduction

We possess strong intuition for how various tuned hyperparameters, such as learning rate or weight decay, affect model training dynamics and outcomes (Galanti et al., 2023; Lyu et al., 2022). For example, a larger learning rate may lead to faster convergence at the cost of sub-optimal solutions (Hazan, 2019; Smith et al., 2021; Wu et al., 2019). However, we lack similar intuitions for the impact of randomness. Like other hyperparameters, random seeds also have a significant impact on training (Madhyastha & Jain, 2019; Sellam et al., 2022), but we have a limited understanding of how randomness in training actually manifests in the model.

In this work, we study the impact of random seeds through a low-dimensional representation of training dynamics, which we use to visualize and cluster training trajectories with different parameter initializations and data orders. Specifically, we analyze training trajectories using a **hidden Markov model** (HMM) fitted on a set of generic metrics collected throughout training, such as the means and variances of the

neural network's weights and biases. From the HMM, we derive a visual summary of how learning occurs for a task across different random seeds.

This work is a first step towards a principled and automated framework for understanding variation in model training. By learning a low-dimensional representation of training trajectories, we analyze training at a higher level of abstraction than directly studying model weights. We use the HMM to infer a Markov chain over latent states in training and relate the resulting paths through the Markov chain to training outcomes.

Our contributions:

1. We propose to use the HMM as a principled, automated, and efficient method for analyzing variability in model training. We fit the HMM to a set of off-the-shelf metrics and allow the model to infer latent state transitions from the metrics. We then extract from the HMM a "training map," which visualizes how training evolves and describes the important metrics for each latent state (Section 2).

   To show the wide applicability of our method, we train HMMs on training trajectories derived from grokking tasks, language modeling, and image classification across a variety of model architectures and sizes. In these settings, we use the training map to characterize how different random seeds lead to different training trajectories. Furthermore, we analyze **phase transitions** in grokking by matching them to their corresponding latent states in the training map, and thus the important metrics associated with each phase transition (Section 3.1).

2. We discover **detour** states, which are learned latent states associated with slower convergence. We identify detour states using linear regression over the training map and propose our regression method as a general way to assign semantics onto latent states (Sections 2.3, 3.4).

   To connect detour states to optimization, we discover that we can induce detour states in image classification by destabilizing the optimization process and, conversely, remove detour states in grokking by stabilizing the optimization process. By making a few changes that are known to stabilize neural network training, such adding normalization layers, we find that the gap between memorization and generalization in grokking is dramatically reduced. Our results, along with prior work from Liu et al. (2023), show that grokking can be avoided by changing the architecture or optimization of deep networks (Section 3.3).

Our code is available at `https://github.com/michahu/modeling-training`.

## 2 Methods

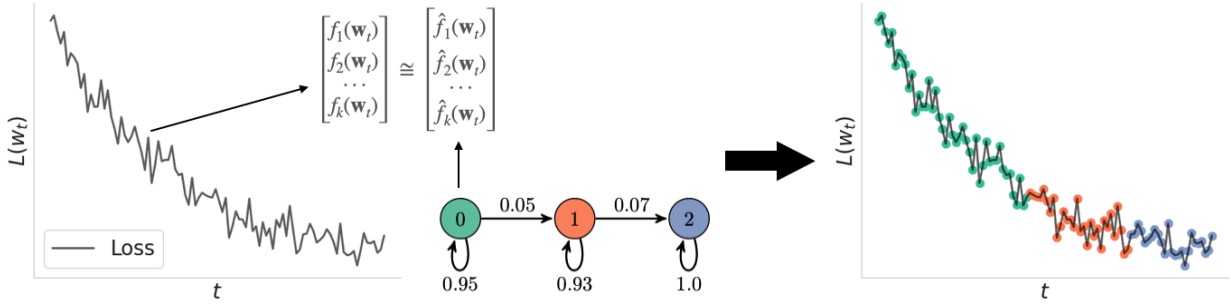

Figure 1: From training runs we collect metrics, which are functions of the neural networks' weights. We then train a hidden Markov model to predict the sequences of metrics generated from the training runs. The hidden Markov model learns a discrete latent state over the sequence, which we use to cluster and analyze the training trajectory.

In this work, we cluster training trajectories from different random seeds and then analyze these clusters to better understand their learning dynamics and how they compare to each other. To cluster trajectories, we assign each model checkpoint to a discrete latent state using an HMM. We choose the HMM because it is a

simple time series model with a discrete latent space, and we specifically pick a discrete latent space because previous works (Nanda et al., 2023; Olsson et al., 2022) have shown that learning can exhibit a few discrete, qualitatively distinct states.

Let $\mathbf{w}_{1:T} \in \mathbb{R}^{D \times T}$ be the sequence of neural network weights observed during training. Each $\mathbf{w}_t$ is a model checkpoint. In this work, we use the Gaussian HMM to label each checkpoint $\mathbf{w}_{1:T}$ with its own latent state, $s_{1:T}$. Fitting the HMM directly over the weights is computationally infeasible, because the sample complexity of an HMM with $O(D^2)$ parameters would be prohibitively high. Our solution to this problem is to compute a small number of metrics $f_1(\mathbf{w}_{1:T}), \ldots, f_d(\mathbf{w}_{1:T})$ from $\mathbf{w}_{1:T}$, where $d \ll D$ and $f_i : \mathbb{R}^D \to \mathbb{R}, 1 \le i \le d$.

## 2.1 Training an HMM over Metrics

In this work, we focus on capturing how the computation of the neural network changes during training by modeling the evolution of the neural network weights. To succinctly represent the weights, we compute various metrics such as the average layer-wise $L_1$ and $L_2$ norm, the mean and variances of the weights and biases in the network, and the means and variances of each weight matrix's singular values. A full list of the 14 metrics we use, along with formulae and rationales, is in Appendix B.

To fit the HMM, we concatenate these metrics into an observation sequence $z_{1:T}$. We then apply z-score normalization (also known as standardization), adjusting each feature to have a mean of zero and a standard deviation of one, as HMMs are sensitive to the scale of features. We thus obtain the normalized sequence $\tilde{z}_{1:T}$. To bound the impact of training trajectory length, we compute z-scores using the estimated mean and variance of (up to) the first 1000 collected checkpoints.

$$
z_t = \begin{bmatrix} f_1(\mathbf{w}_t) \\ \vdots \\ f_d(\mathbf{w}_t) \end{bmatrix}, \qquad \tilde{z}_t = \begin{bmatrix} [f_1(\mathbf{w}_t) - \mu(f_1(\mathbf{w}_{1:T}))]/\sigma(f_1(\mathbf{w}_{1:T})) \\ \vdots \\ [f_d(\mathbf{w}_t) - \mu(f_d(\mathbf{w}_{1:T}))]/\sigma(f_d(\mathbf{w}_{1:T})) \end{bmatrix}
$$

In total, we collect $N$ observation sequences $\{z_{1:T}\}_1^N$ from $N$ different random seeds, normalize the distribution of each metric across training for a given seed, and train the HMM over the sequences $\{\tilde{z}_{1:T}\}_1^N$ using the Baum-Welch algorithm (Baum et al., 1970). The main hyperparameter in the HMM is the number of hidden states, which is typically tuned using the log-likelihood, Akaike information criterion (AIC), and/or Bayesian information criterion (BIC). (Akaike, 1998; Schwarz, 1978) Here, we hold out 20% of the $N$ trajectories as validation sequences and choose the number of hidden states that minimizes the BIC. We use BIC because BIC imposes a stronger preference for simpler, and thus more interpretable, models. Model selection curves are in Appendix H.

## 2.2 Extracting the Training Map

Next, we use the HMM to describe the important features of each hidden state and how the hidden states relate to each other. We convert the HMM into a "training map," which represents hidden states as vertices and hidden state transitions as edges in a state diagram (see Figure 2).

First, we extract the state diagram's structure from the HMM. The learned HMM has two sets of parameters: the transition matrix $p(s_t|s_{t-1})$ between hidden states, and the emission distribution $p(\tilde{z}_t|s_t = k) \sim N(\mu_k, \Sigma_k)$, where $\mu_k$ and $\Sigma_k$ are the mean and covariance of the Gaussian conditioned on the hidden state $k$, respectively. The transition matrix is a Markov chain that defines the state diagram's structure: the hidden states and the possible transitions, or edges, between hidden states *a priori*. We set edge weights in the Markov chain to zero if the edge does not appear in any of the $N$ hidden state trajectories inferred by the HMM.

We annotate the hidden states $s_{1:T}$ by ranking the features $\tilde{z}_t[i]$ according to the absolute value of the log posterior's partial derivative with respect to $\tilde{z}_t$:[1]

$$\left| \frac{\partial \log p(s_t = k | \tilde{z}_{1:t})}{\partial \tilde{z}_t} \right|$$

The absolute value of this partial derivative is a vector. Intuitively, if the $i$th index in this vector is large, then changes in the feature $\tilde{z}_t[i]$ strongly influence the prediction that $s_t = k$. We compute the closed form of this derivative in Appendix A.

To characterize edges in the state diagram where the hidden state changes $j \to k$, we use this derivative, along with the learned means $\mu_j$ and $\mu_k$. A hidden state change from $j$ to $k$ means the new observation has moved closer to $\mu_k$. Thus, we can summarize the movement of features from $j$ to $k$ using the difference vector $\mu_k - \mu_j$. However, not all these changes are necessarily important for the belief that $s_t = k$. To account for this, we can rank these changes by our measure of influence, computed from partial derivatives of the posterior.

In the results to follow, when examining a state transition $j \to k$ at timestep $t$, we report the 3 most influential features for hidden state $k$. To aggregate across runs, we average the absolute value vectors.

In summary, we can obtain a training map from an HMM by extracting:

- The state diagram structure from a pruned transition matrix.
- Edge labels from 1) differences between learned means and 2) partial derivatives of the posterior.

### 2.3 Assigning Semantics to Latent States

From the HMM's transition matrix, we obtain a training map, or the Markov chain between learned latent states of training. We then label the transitions in the training map using the HMM's learned means and partial derivatives of the posterior. But what do we learn from the path a training run takes through the map? In particular, what impact does visiting a particular state have on training outcomes?

In order to relate HMM states to training outcomes, we select a metric and predict it from the path a training run takes through the Markov chain. To do so, we must featurize the sequence of latent states, and in this work we use unigram featurization, or a "bag of states" model. Formally, let $s_1, s_2, \ldots, s_T$ be the latent states visited during a training run. The empirical distribution over states can be calculated as:

$$\hat{P}(s = k) = \frac{\sum_j \mathbb{1}(s_j = k)}{T}$$

where $k$ represents a particular state and $T$ is the total number of checkpoints in the trajectory. This distribution can be written as a $d$-dimensional vector, which is equivalent to unigram featurization.

In this work, we investigate how particular states impact convergence time, which we measure as the first timestep that evaluation accuracy crosses a threshold. We set the threshold to be a value slightly smaller than the maximum evaluation accuracy (see Section 3.4). We use linear regression to predict convergence time from $\hat{P}$. Here, we are not forecasting when a model will converge from earlier timesteps; rather, we are simply using linear regression to learn a function between latent states and convergence time.

After training the regression model, we examine the regression coefficients to see which states are correlated with slower or faster convergence times. If the regression coefficient for a state is positive when predicting convergence time, then a training run spending additional time in that state implies longer convergence time. Additionally, if that same state is not visited by all trajectories, then we can consider it a **detour**, because the trajectories that visit the optional state are also delaying their convergence time.

---

[1]Computing feature importances using partial derivatives of the posterior was suggested by Nguyen Hung Quang and Khoa Doan. Previous versions of this paper used a different computation method.

**Definition.**  A learned latent state is a **detour state** if:

- Some training runs converge without visiting the state. This indicates that the state is "optional."

- Its linear regression coefficient is positive when predicting convergence time. This indicates that a training run spending more time in the state will have a longer convergence time.

Our method for assigning semantics to latent states can be extended to other metrics. For example, one might use regression to predict a measure of gender bias, which can vary widely across training runs (Sellam et al., 2022), from the empirical distribution over latent states. The training map then becomes a map of how gender bias manifests across training runs. We recommend computing the $p$-value of the linear regression and only interpreting the coefficients when they are statistically significant.

## 3   Results

To show the applicability of our HMM-based method across a variety of training settings and model architectures, we perform experiments across five tasks: modular addition, sparse parities, masked language modeling, MNIST, and CIFAR-100. For all hyperparameter details, see Appendix D. In this work, we ignore embedding matrices and layer norms when computing metrics, as we are primarily interested in how the function represented by the neural network changes.

Modular arithmetic and sparse parities are tasks where models consistently exhibit **grokking** (Power et al., 2022), a phenomenon where the training and validation losses seem to be decoupled, and the validation loss drops sharply after a period of little to no improvement. The model first memorizes the training data and then generalizes to the validation set. We call these sharp changes "phase transitions," which are periods in training which contain an inflection in the loss (i.e., the concavity of the loss changes) that is then sustained (no return to chance performance).

We study modular arithmetic and sparse parities to see how phase transitions are represented by the HMM's discrete latent space. We complement these tasks with masked language modeling (Appendix E) and image classification. In Sections 3.1 and 3.2, we use the training map to examine the characteristics of slow and fast-converging training runs in the grokking settings and image classification. In Section 3.3, we show that variation in convergence times between runs can be modulated by changing training hyperparameters or model architecture. Finally, in Section 3.4 we formalize the observations of Section 3.3, using linear regression to connect detour states with convergence time.

### 3.1   Algorithmic Data: Modular Arithmetic and Sparse Parities

**Modular Arithmetic: Figure 2.**  In modular addition, we train a one-layer autoregressive transformer to predict $z = (x + y) \mod 113$ from inputs $x$ and $y$. We collect trajectories using 40 random seeds and train and validate the HMM on a random 80-20 validation split, a split that we use for all settings. This is a replication of the experiments in Nanda et al. (2023).

In modular arithmetic, some training runs converge thousands of epochs earlier than others. Examining the modular addition training map, we find several paths of different lengths: some training runs take the shortest path through the map to convergence, while others do not. We feature three such paths in Figure 2. All runs initialize in state 1 and achieve low loss in state 3, but there are several paths from 1 to 3. The longest path $(1 \to 5 \to 2 \to 3)$ coincides with the longest time to convergence of the three featured runs, and the shortest path $(1 \to 3)$ with the shortest.

Using the HMM, we further dissect this variability by relating the edges exiting state 1 to how fast or slow generalizing runs differ with respect to model internals. The results of this examination are in the table of Figure 2. Here, we take the top 3 features of states 2, 5, and 3 via the learned covariance matrices, and quantify the feature movements of the top 3 features by subtracting the learned means (recall $\tilde{z}$) between these states and state 1. We find that the fast-generalizing path $(1 \to 3)$ is characterized by a "just-right"

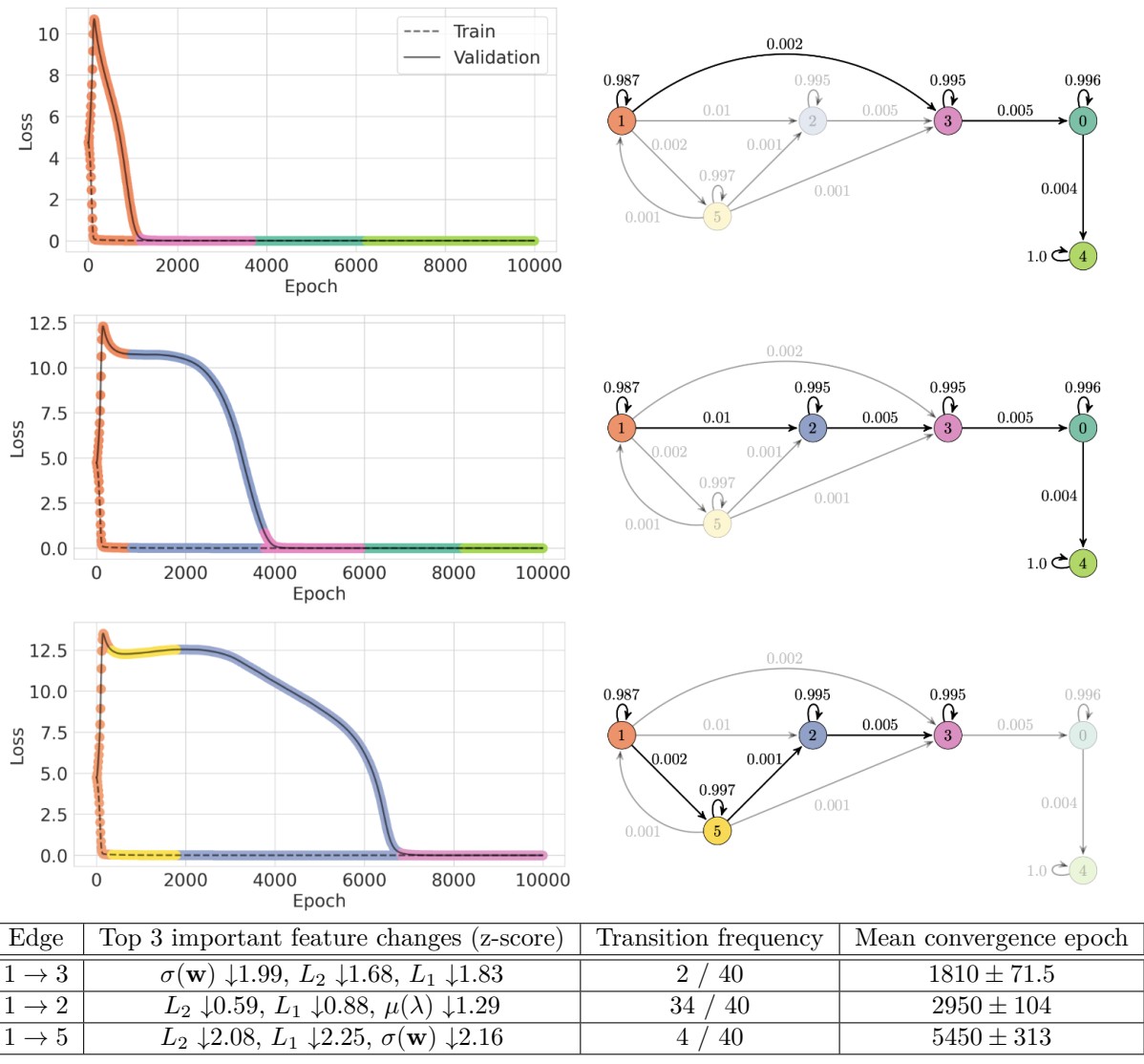

| Edge | Top 3 important feature changes (z-score) | Transition frequency | Mean convergence epoch |
|------|-------------------------------------------|----------------------|------------------------|
| $1 \rightarrow 3$ | $\sigma(\mathbf{w}) \downarrow 1.99$, $L_2 \downarrow 1.68$, $L_1 \downarrow 1.83$ | 2 / 40 | $1810 \pm 71.5$ |
| $1 \rightarrow 2$ | $L_2 \downarrow 0.59$, $L_1 \downarrow 0.88$, $\mu(\lambda) \downarrow 1.29$ | 34 / 40 | $2950 \pm 104$ |
| $1 \rightarrow 5$ | $L_2 \downarrow 2.08$, $L_1 \downarrow 2.25$, $\sigma(\mathbf{w}) \downarrow 2.16$ | 4 / 40 | $5450 \pm 313$ |

Figure 2: One-layer transformer trained on modular addition. Edges exiting the initialization state 1 all have different mean convergence epochs. Feature changes are ordered by importance from most to least. For example, "$L_2 \downarrow 0.59$" means that state 2 has a learned $L_2$ norm that is 0.59 standard deviations lower than state 1, and the $L_2$ norm is the most important feature for state 2. See Appendix B for a glossary of metrics and Section 2.2 for how we identify important features.

drop in the $L_2$ norm ($\downarrow 1.68$, see table). The slower-generalizing runs ($1 \rightarrow 2 \rightarrow 3$) and ($1 \rightarrow 5 \rightarrow 2 \rightarrow 3$) are characterized by either smaller ($\downarrow 0.59$) or larger ($\downarrow 2.08$) drops in $L_2$ norm.

We can also connect our training map results to phase transitions found in modular addition by prior work Nanda et al. (2023); Power et al. (2022): State 1 encapsulates the memorization phase transition: the training loss drop to near-zero in state 1, while validation loss increases. Thus, according to the training map, the epoch in which the generalization phase transition happens is affected by how fast the $L_2$ norm drops immediately after the memorization phase transition. A "just-right" drop in the $L_2$ norm is correlated with the quickest onset of generalization.

**Sparse Parities: Figure 8 in Appendix F.** Sparse parities is a similar rule-based task to modular addition, where a multilayer perceptron must learn to apply an $AND$ operation to 3 bits within a 40-length bit vector; the crux of the task is learning which 3 of the 40 bits are relevant. We again collect 40 training runs.

Similar to modular arithmetic, path variability through the training map also appears at the beginning of training in sparse parities. Slow-generalizing runs take the path $(2 \rightarrow 0 \rightarrow 5)$, while fast-generalizing runs take the more direct path $(2 \rightarrow 5)$. The $L_2$ norm remains important here, with the edge $(2 \rightarrow 0)$ characterized by an increase in the $L_2$ norm and the edge $(2 \rightarrow 5)$ characterized by a decrease. Once again, the speed at which the generalization phase transition occurs is associated with a specific change in the $L_2$ norm immediately after the memorization phase transition.

## 3.2 Image classification: CIFAR-100 and MNIST

**CIFAR-100: Figure 3** Training neural networks on algorithmic data is a nascent task. As a counterpoint to the grokking settings, consider image classification, a well-studied task in computer vision and machine learning. We collect 40 runs of ResNet18 (He et al., 2016) trained on CIFAR-100 (Krizhevsky, 2009), and find that the learning dynamics are smooth and insensitive to random seed. Unlike our results from the prior section, the training map for CIFAR-100 is a linear graph, and the state transitions all tend to feature increasing dispersion in the weights. We show the top 3 features for each state transition in the table of Figure 3. The $L_1$ and $L_2$ norms are increasing monotonically across all state transitions.

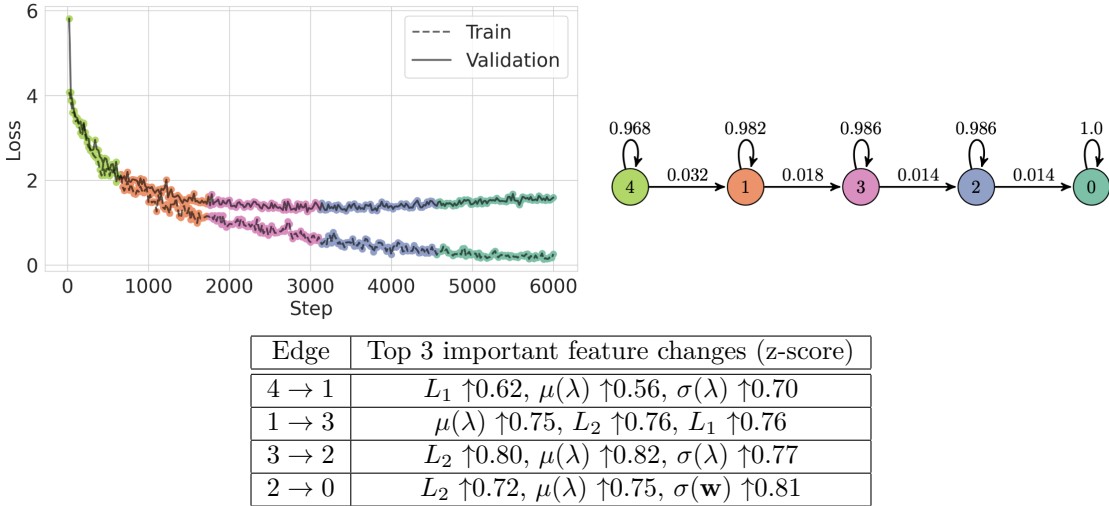

| Edge | Top 3 important feature changes (z-score) |
|---|---|
| $4 \rightarrow 1$ | $L_1 \uparrow 0.62$, $\mu(\lambda) \uparrow 0.56$, $\sigma(\lambda) \uparrow 0.70$ |
| $1 \rightarrow 3$ | $\mu(\lambda) \uparrow 0.75$, $L_2 \uparrow 0.76$, $L_1 \uparrow 0.76$ |
| $3 \rightarrow 2$ | $L_2 \uparrow 0.80$, $\mu(\lambda) \uparrow 0.82$, $\sigma(\lambda) \uparrow 0.77$ |
| $2 \rightarrow 0$ | $L_2 \uparrow 0.72$, $\mu(\lambda) \uparrow 0.75$, $\sigma(\mathbf{w}) \uparrow 0.81$ |

Figure 3: ResNet18 trained on CIFAR-100. All 40 training runs we collected from CIFAR-100 follow the same path, although individual runs can spend slightly different amounts of time in each state. As shown by the table, the training dynamics of CIFAR-100 are similar between states.

**MNIST: Figure 9 in Appendix G.** The dynamics of MNIST are similar to that of CIFAR-100. We collect 40 training runs of a two-layer MLP learning image classification on MNIST, with hyperparameters based on Simard et al. (2003). The training runs of MNIST again follow a single trajectory through the training map. We examine several state transitions throughout training and find that the transitions are also characterized by monotonically increasing changes between features.

## 3.3 Destabilizing Image Classification, Stabilizing Grokking

From the previous two sections, we observe that the training dynamics of neural networks learning algorithmic data (modular addition and sparse parities) are highly sensitive to random seed, while the dynamics of networks trained on image classification are relatively unaffected by random seed. We will now show that

this difference in random seed sensitivity is due to hyperparameter and model architecture decisions within the training setups that we chose to replicate. Variability in training dynamics is not a necessarily a feature of the task, and it is not a feature of the tasks we examine in this paper. Grokking is also affected by model architecture and optimization hyperparameters, and small changes to training can both close the gap between memorization and generalization in grokking and make training robust to changes in random seed. Furthermore, removing improvements to the image classification training process can induce variability in training where it previously did not exist.

First, we examine the training dynamics of ResNets without batch normalization (Ioffe & Szegedy, 2015) and residual connections. Residual connections help ResNets avoid vanishing gradients (He et al., 2016) and smooth the loss landscape (Li et al., 2018). Batch norm has similarly been shown to add smoothness to the loss landscape (Santurkar et al., 2018) and also contributes to automatic learning rate tuning (Arora et al., 2019). We remove batch norm and residual connections from ResNet18 and train the ablated networks from scratch on CIFAR-100 over 40 random seeds.

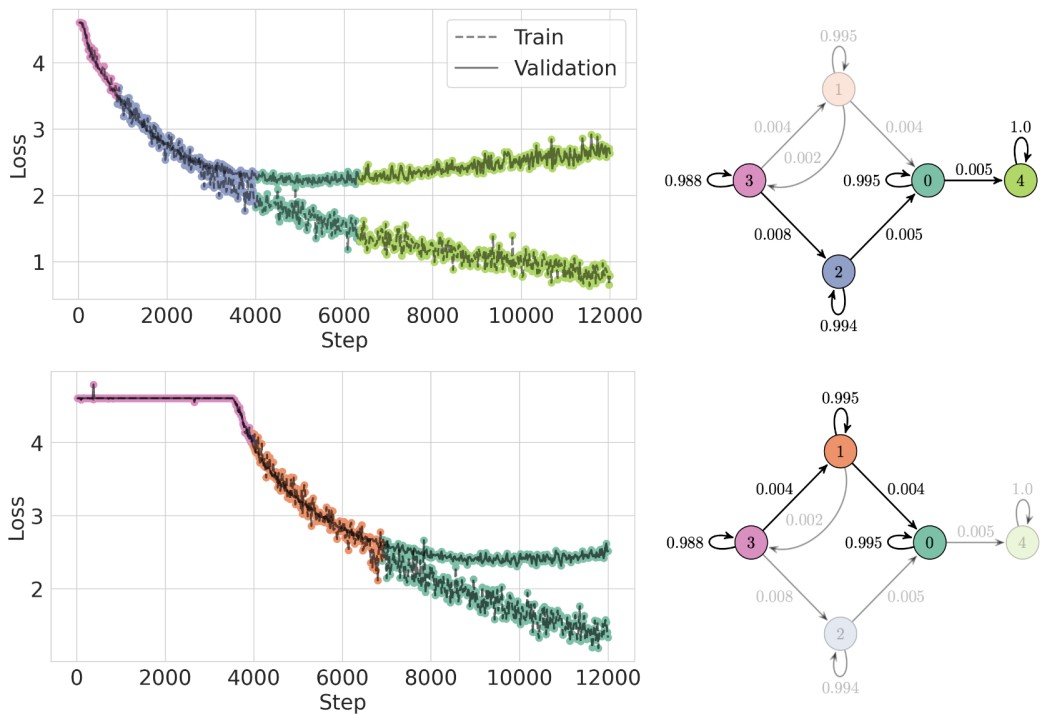

| Edge | Top 3 important feature changes (z-score) | Transition frequency | Mean convergence step |
|------|-------------------------------------------|----------------------|-----------------------|
| $3 \rightarrow 2$ | $\mu(\lambda) \downarrow 0.17$, $L_1 \downarrow 0.18$, $\frac{L_1}{L_2} \downarrow 2.12$ | 29 / 40 | $3530 \pm 460$ |
| $3 \rightarrow 1$ | $\mu(\lambda) \uparrow 0.67$, $L_2 \uparrow 0.74$, $L_1 \uparrow 0.63$ | 12 / 40 | $7260 \pm 3010$ |

Figure 4: Without residual connections and batch normalization, ResNet training becomes unstable, causing convergence times to differ significantly. Slow-generalizing runs take the state transition $(3 \rightarrow 1)$, while fast-generalizing runs take the state transition $(3 \rightarrow 2)$. (Runs can take the path $(3 \rightarrow 1 \rightarrow 3 \rightarrow 2)$, so transition frequencies do not sum to 40). The variability induced by removing residual connections and batch norm occurs at the beginning of training.

In this experiment, we show that changing the training dynamics of a task also changes the training map. Without batch norm and residual connections, ResNet18's training dynamics become significantly more sensitive to randomness. See Figure 4. Depending on the random seed, the model may stagnate for many updates before generalizing. This increase in random variation is visible in the learned training map, which now forks when exiting state 3, the initialization state. There now exists a slow-generalizing path $(3 \rightarrow 1)$ and a fast-generalizing path $(3 \rightarrow 2)$, characterized by feature movements in opposite directions.

If removing batch normalization destabilizes ResNet training in CIFAR-100, then adding layer normalization (which was removed by Nanda et al. (2023)) should stabilize training in modular addition. Thus, we add layer normalization back in and train over 40 random seeds. We also decrease the batch size, which leads SGD to flatter minima (Keskar et al., 2017). These modifications to training help the transformer converge around 30 times faster on modular addition data. Furthermore, sensitivity to random seed disappears–the training map for modular addition in Figure 5 becomes a linear graph.

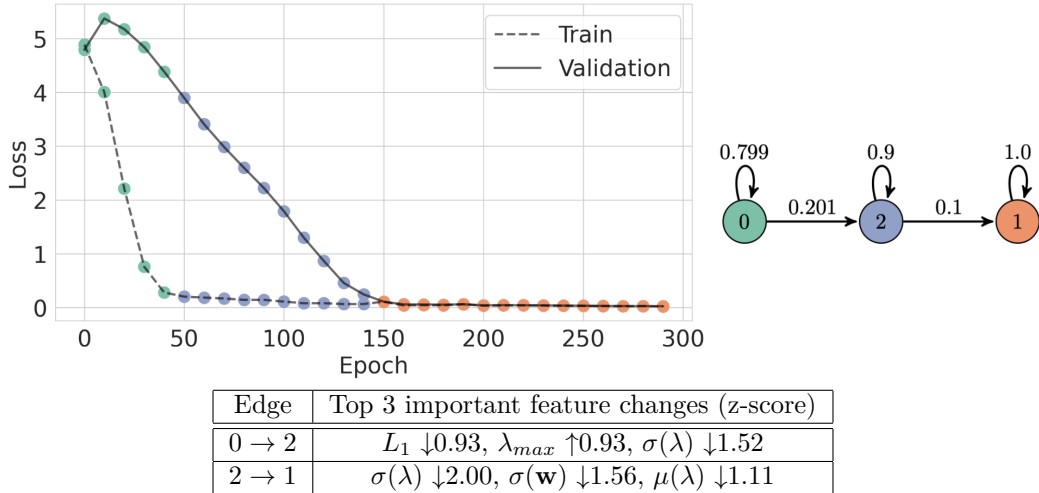

| Edge | Top 3 important feature changes (z-score) |
|---|---|
| $0 \rightarrow 2$ | $L_1 \downarrow 0.93$, $\lambda_{max} \uparrow 0.93$, $\sigma(\lambda) \downarrow 1.52$ |
| $2 \rightarrow 1$ | $\sigma(\lambda) \downarrow 2.00$, $\sigma(\mathbf{w}) \downarrow 1.56$, $\mu(\lambda) \downarrow 1.11$ |

Figure 5: With layer normalization and a lower learning rate, the one-layer transformer quickly learns the modular arithmetic task, with a convergence time stable across random seed. This stability is captured by the linear training map. Critically, the map still reflects the grokking phase transitions: memorization, which occurs in state 0, and generalization, which occurs in state 2.

From this section, we draw two conclusions. First, that model training choices can amplify or minimize the grokking effect. Second, that using different hyperparameters or architectures can result in different training maps for the same task. In training setups sensitive to random seed, the HMM associates differences in training dynamics with different latent states.

### 3.4 Predicting Convergence Time

In Section 3.1, we identified latent states visited by slow-generalizing runs that were skipped by fast-generalizing runs. We now use our framework for assigning semantics to latent states from Section 2.3 to identify these skipped latent states as detour states, or states that slow down convergence. The first step in our framework is to use paths through the training map as features in a linear regression to predict convergence time. We define convergence time as the iteration where validation accuracy is greater than some threshold, and we take this threshold to be 0.9 in modular addition and sparse parities, 0.6 for the stable version of CIFAR-100, 0.4 for destabilized CIFAR-100, and 0.97 for MNIST. We set these values to be slightly less than the maximum evaluation accuracy for each task, respectively. To visualize the variance in convergence times, see Appendix I.

In Table 1, we find that linear regression predicts convergence time from a given training run's empirical distribution over latent states very accurately, as long as the training map contains forked paths. If the training map is instead linear, training follows similar paths through the HMM across different random seeds. We formalize this intuition of **trajectory dissimilarity** as the expected Wasserstein distance $W(\cdot, \cdot)$ (Kantorovich, 1939; Vaserstein, 1969) between any two empirical distributions $p$ and $q$, sampled uniformly over the $N$ random seeds.

$$\text{Trajectory dissimilarity} := \mathbb{E}[W(p,q)] = \frac{2}{N(N-1)} \sum_{i=1}^{N} \sum_{j=1}^{i} W(p_i, q_j) \tag{1}$$

| Dataset | $R^2$ | $p$-value | Dissimilarity | Forking |
|---|---|---|---|---|
| Modular addition | 0.977 | <0.001 | 0.496 | ✓ |
| Modular addition, stabilized | 0.514 | <0.001 | 0.038 | |
| CIFAR-100 | 0.094 | 0.469 | 0.028 | |
| CIFAR-100, destabilized | 0.905 | <0.001 | 0.806 | ✓ |
| Sparse parities | 0.961 | <0.001 | 0.183 | ✓ |
| MNIST | 0.049 | 0.611 | 0.063 | |

Table 1: Predictability of convergence epoch using a unigram model of states. Dissimilarity is provided per Equation 1 and the training maps are marked as forking unless they are linear.

With statistically significant ($p < 0.001$) regression models for modular addition, sparse parities, and destabilized CIFAR-100, we can use the learned regression coefficients to find detour states. In Table 2, we highlight these detour states, defined as any state with a positive regression coefficient that is only visited by a strict subset of training trajectories. In our tasks with linear graphs, there are no detour states, because every training run visits every latent state. Our regression analysis largely confirms observations drawn from looking at the training maps and trajectories in sections prior: states 2 and 5 are detour states in modular addition, state 0 is a detour state in sparse parities, and state 1 is a detour state in destabilized CIFAR-100.

| State | Coefficient |
|---|---|
| 0 | -0.15 |
| 1 | 0.98 |
| 2 | **1.19** |
| 3 | -0.20 |
| 4 | 0.18 |
| 5 | **0.95** |

(a) Modular addition

| State | Coefficient |
|---|---|
| 0 | **0.77** |
| 1 | 0.41 |
| 2 | 0.98 |
| 3 | -0.23 |
| 4 | 0.58 |
| 5 | 1.13 |

(b) Sparse parities

| State | Coefficient |
|---|---|
| 0 | 0.66 |
| 1 | **1.20** |
| 2 | 0.28 |
| 3 | 1.91 |
| 4 | 1.12 |

(c) CIFAR-100, destabilized

Table 2: Learned linear regression coefficients. If the value is positive, then the time spent in the state is correlated with increased convergence time, and vice versa. Detour states are bolded.

Detour states signal that the outcome of training is unstable: they appear in training setups that are sensitive to randomness, and they disappear in setups that are robust to randomness. By adding layer norm and decreasing batch size, we remove detour states in modular addition, and the training map becomes a linear graph. Conversely, removing batch norm and residual connections destabilizes the training of ResNets, thereby inducing forks in the training map that lead to detour states.

## 4 Related Work

Prior works have examined the effect of random seed on training outcome (Sellam et al., 2022; Picard, 2023; Fellicious et al., 2020). To our knowledge, this is the first work to 1) analyze random seed using a probabilistic model and 2) show how random seed manifests as specific changes in metrics during training. Weiss et al. (2018; 2019) model the computation of neural networks as deterministic finite automata (DFA), which bears some similarity to the annotated Markov chain we extract from training runs. Williams (1992) use an extended Kalman filter (EKF) to train a recurrent neural network and note the similarity between EKF and the real-time recurrent learning algorithm (Marschall et al., 2020). In contrast to the existing literature, we use state machines to understand the training process rather than the inference process. Measuring the state of a neural network using various metrics was also done in Frankle et al. (2020).

Analyzing time series data using a probabilistic framework has been successfully applied to many other tasks in machine learning (Kim et al., 2017; Hughey & Krogh, 1996; Bartolucci et al., 2014). In a similar spirit to our work, Batty et al. (2019) use an autoregressive HMM (ARHMM) to segment behavioral videos

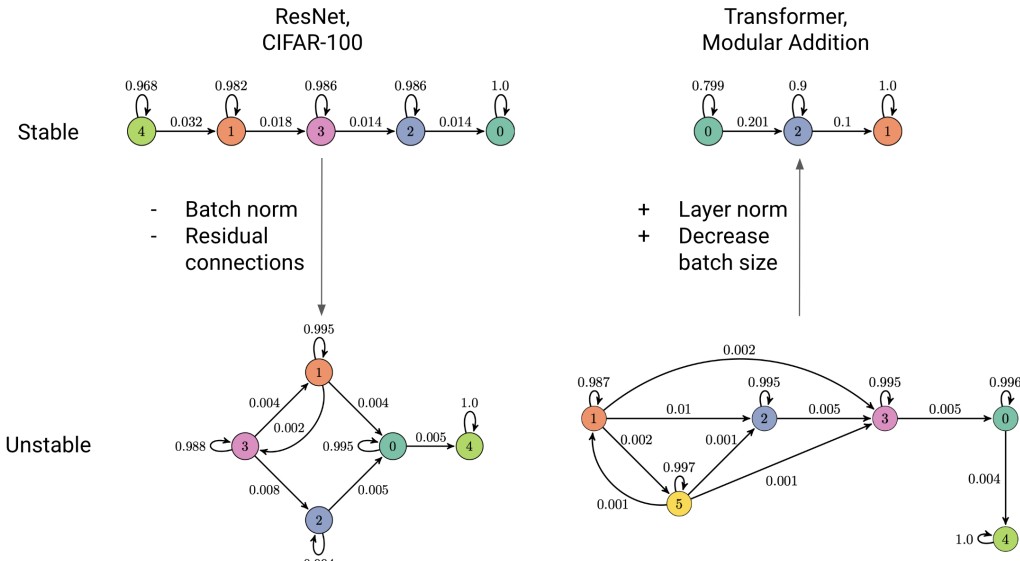

Figure 6: Training maps express variability in training dynamics as a more densely connected graph. For stable training setups, the HMM learns a linear graph as the training map. Training dynamics can be stabilized or destabilized by changing hyperparameters (batch size) or architecture (normalization layers, residual connections).

into semantically similar chunks. The ARHMM can capture both discrete and continuous latent dynamics, making it an interesting model to try for future work.

Our work is substantively inspired by the progress measures literature, which aims to find metrics that can predict discontinuous improvement or convergence in neural networks. Barak et al. (2022) first hypothesized the existence of hidden progress measures. Olsson et al. (2022) found a progress measure for induction heads in Transformer-based language models, and Nanda et al. (2023) found a progress measure for grokking in the modular arithmetic task.

The $L_2$ norm is also known to be both important to and predictive of grokking, thereby motivating the use of weight decay to accelerate convergence in grokking settings (Nanda et al., 2023; Power et al., 2022; Thilak et al., 2022). Liu et al. (2023) highlight the importance of the $L_2$ norm by correcting for grokking via projected gradient descent within a fixed-size $L_2$ ball; conversely, they also induce grokking on new datasets by choosing a disadvantageous $L_2$ norm. Our results show that grokking has other available remedies, beyond ones that directly manipulate the $L_2$ norm. Merrill et al. (2023) and Nanda et al. (2023) show that grokking in sparse parities and modular arithmetic (respectively) can be explained by the emergence of a sparse subnetwork within the larger network.

Finally, this work relates broadly to the empirical study of training dynamics. Much of the literature treats learning as a process where increases in training data lead to predictable increases in test performance (Kaplan et al., 2020; Razeghi et al., 2022) and in model complexity (Choshen et al., 2022; Mangalam & Prabhu, 2019; Nakkiran et al., 2019). However, this treatment of training ignores how heterogeneous the factors of training can be. Different capabilities are learned at different rates (Srivastava et al., 2022), different layers converge at different rates (Raghu et al., 2017), and different latent dimensions emerge at different rates (Jarvis et al., 2023; Saxe et al., 2019). While early stages in training can be modeled nearly exactly through simple methods (Hu et al., 2020; Jacot et al., 2018), early stages are notably distinct from later stages, and simple models can often belie common training phenomena (Fort et al., 2020). Consequently, methods like ours which treat training as a heterogeneous process are crucial in understanding realistic training trajectories.

# 5 Discussion

The training maps derived from HMMs are interpretable descriptions of training dynamics that summarize similarities and differences between training runs. Our results show that there exists a low-dimensional, *discrete* representation of training dynamics. Via the HMM, this representation is generally predictive of the next set of metrics in the training trajectory, given the previous metrics. Furthermore, in some cases this low-dimensional, discrete representation can even be used to predict the iteration in which models converge.

## 5.1 Grokking and the Optimization Landscape

We conjecture that grokking is the consequence of a sharp optimization landscape. Consider the edits we performed to significantly decrease the grokking effect: adding layer normalization and decreasing batch size. Normalization layers and decreasing batch size have been documented in the literature as increasing smoothness in the loss landscape (Santurkar et al., 2018; Arora et al., 2019; Keskar et al., 2017). Image classification is a well-studied task with many tricks for improving the efficiency of training; perhaps learning algorithmic data will become just as efficient in the future, such that grokking is no longer a concern.

## 5.2 Progress Measures and Phase Transitions

By modeling convergence time in grokking settings, we analyze phase transitions. We find that the generalization phase transition can be sped up by avoiding detour states. These detour states are generally characterized by specific requirements in metrics such as the $L_2$ norm. For example, in the modular arithmetic setting, avoiding detour states without changing the training setup requires a "just-right" decrease in the $L_2$ norm–not too little, and not too much. This observation aligns with the hypothesis from Liu et al. (2023), where the authors posit that grokking occurs because the weight norm is slow to reach a shell of particular $L_2$ norm in weight space, previously called the "Goldilocks zone" (Fort & Scherlis, 2018).

Our automated approach can be a complement to the progress measures literature, which in previous works has found measures predictive of phase transitions by hand (Barak et al., 2022; Nanda et al., 2023). In this work, instead of carefully choosing a single metric, we compute a variety of metrics and use unsupervised learning to find structure amongst them. We then use the learned latent representation to analyze phase transitions.

## 5.3 The Impact of Random Seed

We recommend that researchers studying training dynamics experiment with a large number of training seeds. When claims are based on a small number of runs, anomalous training phenomena might be missed, simply due to sampling. These anomalous phenomena can be the most elucidating, as in our grokking experiments, where a small number of runs converge faster than the rest. The role of random variation has been highlighted for the performance and generalization of trained models (McCoy et al., 2020; Sellam et al., 2022; Juneja et al., 2023), but there are fewer studies on variation in training dynamics. We recommend studying training across many runs, and possibly relying on state diagrams like ours to distinguish typical and anomalous training phenomena.

## 5.4 Limitations and Future Work

Our work assumes that training dynamics can be represented by a linear, discrete, and Markovian model. Despite the successes of our approach, a higher-powered model might capture even more information about training dynamics. Relaxing the assumptions of the HMM is likely a fruitful area for future work. Additionally, in this work we perform dimensionality reduction via hand-picked metrics. We use these metrics as interpretable features for our training maps, but a fully unsupervised approach without explicit metrics also deserves exploration. For very large models, training an HMM across many random seeds may be infeasible. A possible follow-up work could look at whether models of training dynamics can generalize zero-shot across architectures and architecture sizes (Yang et al., 2021). If this were the case, then one could reuse dynamics models to interpret training.

Finally, our findings are suggestive for future work on hyperparameter search. We demonstrate that 1) training instability to random seed is highly dependent on hyperparameters, and 2) instability manifests early in training. Thus, it may be more efficient to measure early variation across a few seeds to quickly evaluate a hyperparameter setting, rather than waiting to measure the final evaluation accuracy on the trained model.

## 6 Conclusion

We make two main contributions. First, we propose directly modeling training dynamics as a new avenue for interpretability and training dynamics research. We show that even with a simple model like the HMM, we can learn representations of training dynamics that are predictive of key metrics like convergence time. Second, we discover detour states of learning, and show that detour states are related to both how quickly models converge and how sensitive the overall training process is to random seed. Detour states can be removed by finding more efficient training hyperparameters or model architectures.

### Acknowledgements

We would like to thank Nguyen Hung Quang, Khoa Doan, and William Merrill for their insightful comments and suggestions. We particularly thank Quang, who generously reached out about a significant error in a previous version (see Section 2.2) and whose suggested fix has been incorporated into the paper. MYH is supported by an NSF Graduate Research Fellowship. This work was supported by Hyundai Motor Company (under the project Uncertainty in Neural Sequence Modeling), the Samsung Advanced Institute of Technology (under the project Next Generation Deep Learning: From Pattern Recognition to AI), and the National Science Foundation (under NSF Award 1922658).

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

## A    Derivation

We use the log posterior because it has a simplified form for Gaussians. The log posterior is:

$$p(s_t = k | \tilde{z}_{1:t}) = \frac{p(\tilde{z}_t | s_t = k) p(s_t = k, \tilde{z}_{1:t-1})}{p(\tilde{z}_{1:t})}$$

$$\Rightarrow \log p(s_t = k | \tilde{z}_{1:t}) = \log p(\tilde{z}_t | s_t = k) + \log p(s_t = k, \tilde{z}_{1:t-1}) - \log p(\tilde{z}_{1:t})$$

We take the derivative of these three terms separately:

$$\frac{\partial \log p(\tilde{z}_t | s_t = k)}{\partial \tilde{z}_t} = \Sigma_k^{-1} (\mu_k - z_t)$$

$$\frac{\partial \log p(s_t = k, \tilde{z}_{1:t-1})}{\partial \tilde{z}_t} = 0$$

$$\begin{aligned}
\frac{\partial \log p(\tilde{z}_{1:t})}{\partial \tilde{z}_t} &= \frac{1}{p(\tilde{z}_{1:t})} \cdot \frac{\partial p(\tilde{z}_{1:t})}{\partial \tilde{z}_t} \\
&= \frac{1}{p(\tilde{z}_{1:t})} \cdot \frac{\partial \left[ \sum_{h \in \Omega} p(\tilde{z}_t | s_t = h) p(s_t = h, \tilde{z}_{1:t-1}) \right]}{\partial \tilde{z}_t} \\
&= \frac{1}{p(\tilde{z}_{1:t})} \cdot \sum_{h \in \Omega} p(s_t = h, \tilde{z}_{1:t-1}) \frac{\partial p(\tilde{z}_t | s_t = h)}{\partial \tilde{z}_t} \\
&= \frac{1}{p(\tilde{z}_{1:t})} \cdot \sum_{h \in \Omega} p(s_t = h, \tilde{z}_{1:t-1}) p(\tilde{z}_t | s_t = h) \frac{\partial \log p(\tilde{z}_t | s_t = h)}{\partial \tilde{z}_t} \\
&= \sum_{h \in \Omega} \frac{p(s_t = h, \tilde{z}_{1:t-1}) p(\tilde{z}_t | s_t = h)}{p(\tilde{z}_{1:t})} \Sigma_h^{-1} (\mu_h - z_t) \\
&= \sum_{h \in \Omega} p(s_t = h | \tilde{z}_{1:t}) \Sigma_h^{-1} (\mu_h - z_t)
\end{aligned}$$

So, for timestep $t$ and hidden state $k$,

$$\left| \frac{\partial \log p(s_t = k | \tilde{z}_{1:t})}{\partial \tilde{z}_t} \right| = \left| \Sigma_k^{-1} (\mu_k - z_t) - \sum_{h \in \Omega} p(s_t = h | \tilde{z}_{1:t}) \Sigma_h^{-1} (\mu_h - z_t) \right|$$

$p(s_t = h | \tilde{z}_{1:t})$ can be efficiently computed using the forward algorithm. See Jurafsky & Martin (2023), chapter A for a reference.

## B Metrics

The chart in this section lists the 14 statistics we computed for each model checkpoint. We use these statistics to capture either 1) how the neural network weights weights are dispersed in space or the 2) properties of the function computed by a layer. For example, the $L_2$ norm measures dispersion because it describes how far away the weights are from the origin. The spectral norm helps capture the function computed by a neural network because it describes the maximum amount that a vector might change as it passes through a layer.

Of course, 1) and 2) are related, and thus the statistics we compute are also related; the $L_2$ matrix norm upper bounds spectral norm. Our philosophy (and recommendation) is to choose a variety of metrics when modeling training dynamics to allow for interactions between metrics.

For metrics that become infeasible to compute during training at large model sizes, we recommend using streaming algorithms, matrix sketching algorithms, or other approximations such as random projections to make computation more efficient. For example, singular values can be computed using streaming algorithms (Maulik & Mengaldo, 2021; Yu et al., 2017) or on a matrix sketch of reduced size (Ghashami et al., 2016).

| Name | Description |
|---|---|
| 1) $L_1$ | The $L_1$-norm, averaged over matrices. $\frac{1}{K}\|w\|_1 = \frac{1}{K}\sum_{i=1}^{n}|w_i|$, where $K$ is the number of weight matrices in the neural network. We average over matrices so that models with different depths are comparable. |
| 1) $L_2$ | The $L_2$-norm, averaged over matrices. $\frac{1}{K}\|w\|_2 = \frac{1}{K}\sum_{i=1}^{n}\sqrt{w_i^2}$ |
| 1) $\frac{L_1}{L_2}$ | Measures the sparsity of the weights (Repetti et al., 2014). $\frac{1}{K}\sum_{i=1}^{K}\frac{L_1^{(i)}}{L_2^{(i)}}$, which is the metric $\frac{L_1}{L_2}$ averaged over the $K$ weight matrices. Lower is more sparse. For example, a one-hot vector is fully sparse and has code sparsity of 1. See Hurley & Rickard (2008) for a discussion on measures of sparsity. |
| 1) $\mu(\mathbf{w})$ | Sample mean of weight. $\frac{1}{N}\sum_{i=1}^{N}w_i$, where $N$ is the number of parameters in the network. |
| 1) $median(\mathbf{w})$ | Median of the weights, treated as a set concatenated together. |
| 1) $\sigma(\mathbf{w})$ | Sample variance of weights without Bessel's correction. $\frac{\sum_{i=1}^{N}(w_i-\bar{w})^2}{N}$ |
| 1) $\mu(b)$ | Sample mean of the biases. We treat the biases separately because they have a distinct interpretation from the weights. |
| 1) $median(b)$ | Median of the biases, treated as a set concatenated together. |
| 1) $\sigma(b)$ | Sample variance of biases without Bessel's correction. |
| 2) trace | The average trace over $K$ weight matrices. $\frac{1}{K}\sum_{i=1}^{K}\mathtt{tr}(W_k)$, where $W_k$ is the $k$th weight matrix. |
| 2) $\lambda_{max}$ | The average spectral norm. $\frac{1}{K}\sum_{i=1}^{K}\|W_k\|_2$. |
| 2) $\frac{trace}{\lambda_{max}}$ | Average trace over spectral norm. $\frac{1}{K}\sum_{i=1}^{K}\frac{\mathtt{tr}(W_k)}{\|W_k\|_2}$. |
| 2) $\mu(\lambda)$ | Average singular value over all matrices. |
| 2) $\sigma(\lambda)$ | Sample variance of singular values over all matrices. |

Table 3: A glossary of metrics. The "Name" column contains how the metrics appear in the text. The label 1) means the statistic intends to capture how the neural network weights weights are dispersed in space. The label 2) means the statistic intends to capture properties of the function computed by a layer.

## C   Baselines

We compare the performance of the full HMM, trained on all 14 statistics discussed in Appendix B, with two baselines:

1. K-means clustering, which learns a discrete latent space similar to the HMM but does not capture temporal structure.

2. HMM-1, which is the HMM trained to model only the $L_2$ norm. We chose this baseline because we found $L_2$ norm to be one of the most important metrics throughout all settings (see Sections 3.1 and 3.2), and the $L_2$ norm has also been noted as a metric predictive of model qualities in prior works (Liu et al., 2023; Nanda et al., 2023; Thilak et al., 2022).

For each setting, we perform model selection and choose the optimal number of components according to BIC. Below, we list the number of components in the best model, along with its BIC. We find that K-means and HMM-1 tend to use **consistently more components** compared to the base HMM. We consider this undesirable because more components dilutes the interpretation of each individual component. In particular, K-means tends to use the maximum number of clusters we allowed to cluster the given sequence.

(NB: BICs are not comparable across models; we provide them for comparison in case the reader trains a model of the same class.)

| Dataset | k-means | | HMM-1 | | HMM | |
|---|---|---|---|---|---|---|
| | Components | BIC | Components | BIC | Components | BIC |
| Modular | 16 | 103700 | 15 | -14070 | 6 | -5724 |
| Modular, stabilized | 10 | 3864 | 5 | 166.0 | 3 | 759.9 |
| CIFAR-100 | 16 | 19360 | 10 | -1851 | 5 | -124400 |
| CIFAR-100, destabilized | 16 | 23210 | 15 | -1432 | 5 | -59080 |
| Sparse parities | 16 | 23660 | 13 | -2965 | 6 | -49530 |
| MNIST | 16 | 3244 | 11 | -1064 | 6 | -101200 |

Table 4: Dataset Scores

## D   Training Hyperparameters

For the MultiBERTs (Sellam et al., 2022), we use the open-source training checkpoints without any additional training.

| Hyperparameter | Value |
|---|---|
| Learning Rate | 1e-1 |
| Batch Size | 32 |
| Training data size (randomly generated) | 1000 |
| Test data (randomly generated) | 100 |
| Architecture | Multilayer perceptron |
| Number of hidden layers | 1 |
| Model Hidden Size | 128 |
| Weight Decay | 0.01 |
| Seed | 0 through 40 |
| Optimizer | SGD |

Table 5: Sparse parities, replicating Barak et al. (2022)

| Hyperparameter | Value |
|---|---|
| Learning Rate | 1e-3 |
| Batch Size | 2048 |
| Training data size | 3831 (30% of all possible samples) |
| Architecture | Transformer, no layer normalization |
| Transformer Number of Layers | 1 |
| Transformer Number of Heads | 4 |
| Model Hidden Size | 128 |
| Model Head Size | 32 |
| Weight Decay | 1.0 |
| Seed | 0 through 40 |
| Optimizer | AdamW |

Table 6: Modular addition, replicating Nanda et al. (2023). To **stabilize** training (Figure 5), we reduced the batch size from 2048 to 256 and added back layer normalization.

| Hyperparameter | Value |
|---|---|
| Learning Rate | 1e-3 |
| Batch Size | 256 |
| Training data size | 50000 (splits downloaded from PyTorch) |
| Architecture | ResNet18 |
| Weight Decay | 1.0 |
| Seed | 0 through 40 |
| Optimizer | AdamW |
| Data preprocessing | Random crop, random horizontal flip, and normalization |

Table 7: CIFAR-100. To **destabilize** training (Figure 4), we removed batch normalization and residual connections.

| Hyperparameter | Value |
|---|---|
| Learning Rate | 1e-3 |
| Batch Size | 256 |
| Training data size | 60000 (splits downloaded from PyTorch) |
| Architecture | MLP |
| Number of hidden layers | 1 |
| Hidden size | 800 |
| Weight Decay | 1.0 |
| Seed | 0 through 40 |
| Optimizer | AdamW |
| Data preprocessing | Flatten to vector |

Table 8: MNIST. MLP hyperparameters based on Simard et al. (2003).

# E   Language Modeling: MultiBERTs

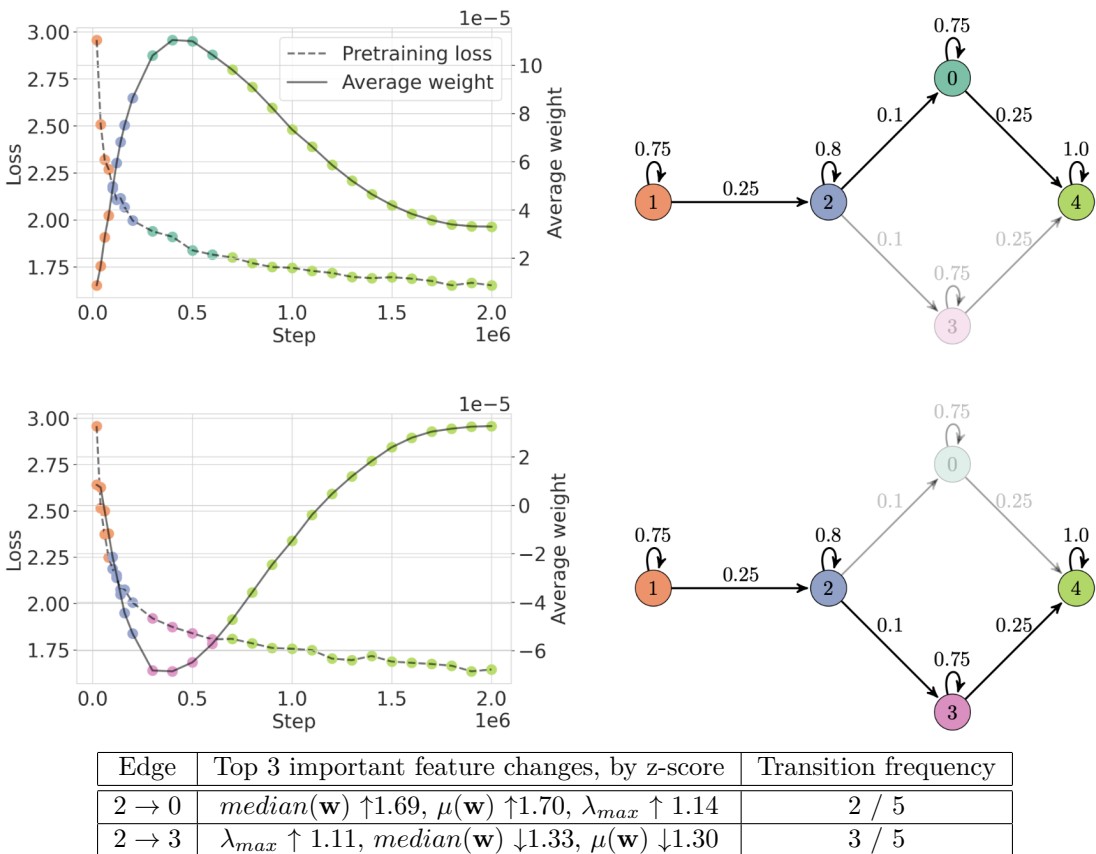

| Edge | Top 3 important feature changes, by z-score | Transition frequency |
|------|---------------------------------------------|----------------------|
| $2 \rightarrow 0$ | $median(\mathbf{w}) \uparrow 1.69, \mu(\mathbf{w}) \uparrow 1.70, \lambda_{max} \uparrow 1.14$ | 2 / 5 |
| $2 \rightarrow 3$ | $\lambda_{max} \uparrow 1.11, median(\mathbf{w}) \downarrow 1.33, \mu(\mathbf{w}) \downarrow 1.30$ | 3 / 5 |

Figure 7: MultiBERTs. The average weight $\frac{1}{N}\sum_i^N w_i$ initially decreases in two of the five MultiBERT runs $(2 \rightarrow 0)$ and increases in the other three $(2 \rightarrow 3)$. However, all runs eventually converge to roughly the same average weight. The HMM represents this difference in runs as the states 0 and 3. Critically, this difference is imperceptible from the pretraining loss.

To study variation in masked language model training, we use the five released training trajectories from the MultiBERTs (Sellam et al., 2022), which are replications of the original BERT model (Devlin et al., 2019), trained under different random seeds. MultiBERTs differs from the other settings we consider because its training occurs over the course of a single epoch, rather than over multiple epochs.

The most notable feature of the MultiBERTs training map is the fork at state 2. The average weights of the MultiBERTs models all converge to around $3.7 \times 10^{-5}$, but the paths that the five models take to get there can be clustered into two different trajectories. For the path including $(2 \rightarrow 0)$, the average weight increases during states 2 and zero and then decreases during state 4, while the opposite is true for paths including $(2 \rightarrow 3)$. Understanding this difference between MultiBERTs models could be a fruitful area for future work. Critically, this difference in model internals is imperceptible from the pretraining loss, which decreases at roughly the same rate for all five MultiBERTs runs. However, the MultiBERTs exhibit significant variation in transfer learning performance and gender bias Sellam et al. (2022), so these paths may indicate differences in behavior under specific distribution shifts and settings.

## F    Algorithmic Data: Sparse Parities

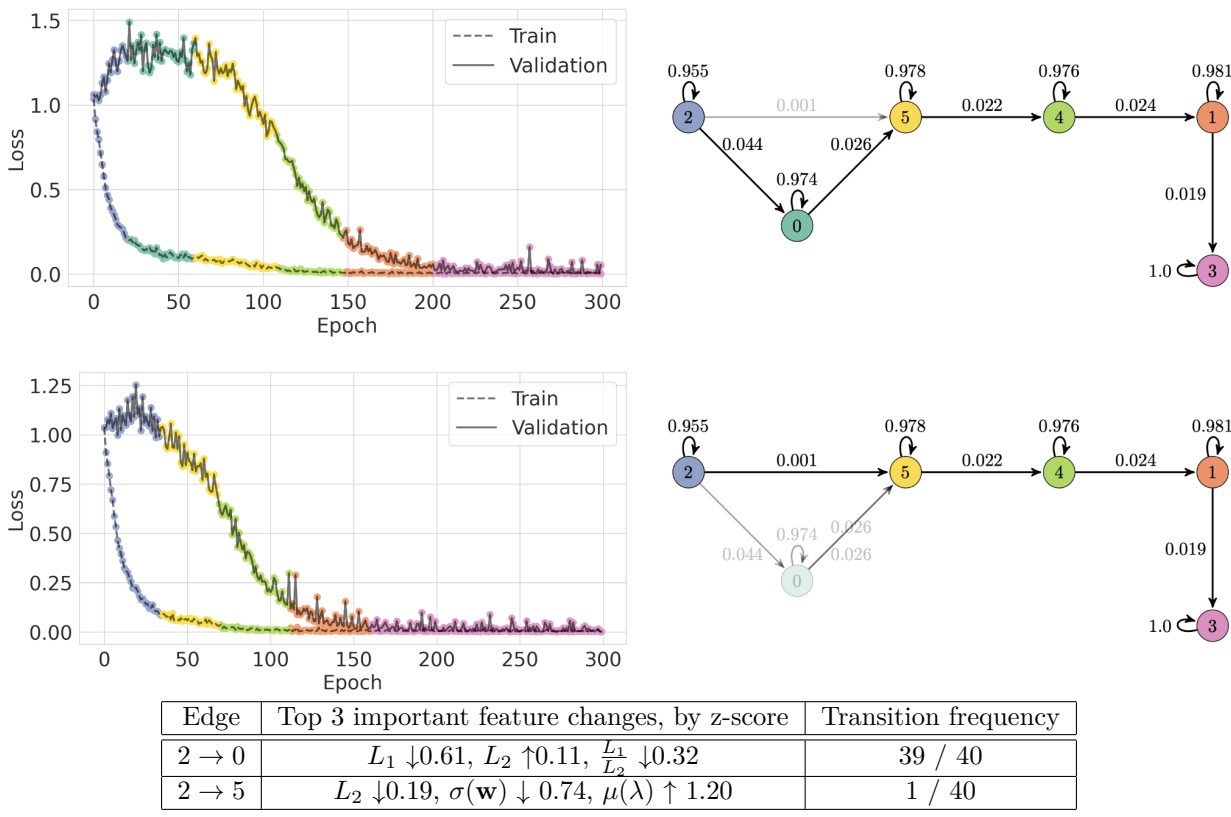

| Edge | Top 3 important feature changes, by z-score | Transition frequency |
|---|---|---|
| $2 \to 0$ | $L_1 \downarrow 0.61$, $L_2 \uparrow 0.11$, $\frac{L_1}{L_2} \downarrow 0.32$ | 39 / 40 |
| $2 \to 5$ | $L_2 \downarrow 0.19$, $\sigma(\mathbf{w}) \downarrow 0.74$, $\mu(\lambda) \uparrow 1.20$ | 1 / 40 |

Figure 8: Sparse parities. Faster generalization in sparse parities occurs with an early decrease in the $L_2$ norm. The norm ratio $\frac{L_1}{L_2}$ is a metric for dispersion, and it decreases as the vector becomes more sparse. For example, a one-hot vector is completely sparse and is the minimum of $\frac{L_1}{L_2}$.

## G  Image Classification: MNIST

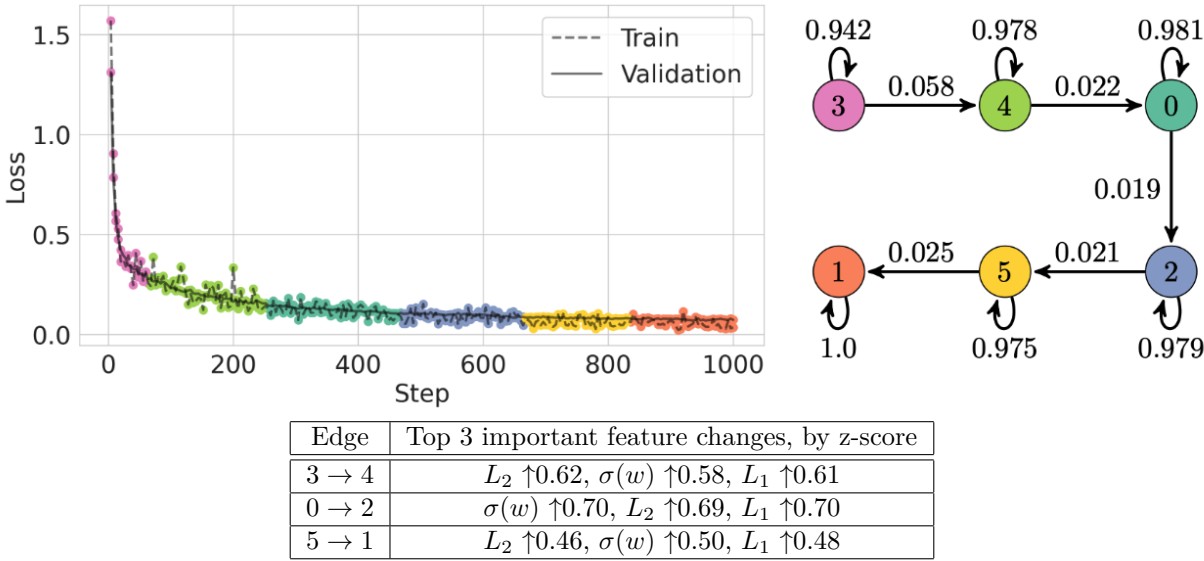

| Edge | Top 3 important feature changes, by z-score |
|------|---------------------------------------------|
| $3 \to 4$ | $L_2$ ↑0.62, $\sigma(w)$ ↑0.58, $L_1$ ↑0.61 |
| $0 \to 2$ | $\sigma(w)$ ↑0.70, $L_2$ ↑0.69, $L_1$ ↑0.70 |
| $5 \to 1$ | $L_2$ ↑0.46, $\sigma(w)$ ↑0.50, $L_1$ ↑0.48 |

Figure 9: MNIST. All 40 training runs we collected from MNIST follow the same path, although individual runs can spend slightly different amounts of time in each state. As shown by the training map and accompanying annotations in the table, the training dynamics of MNIST are similar between states.

## H    Model Selection Curves

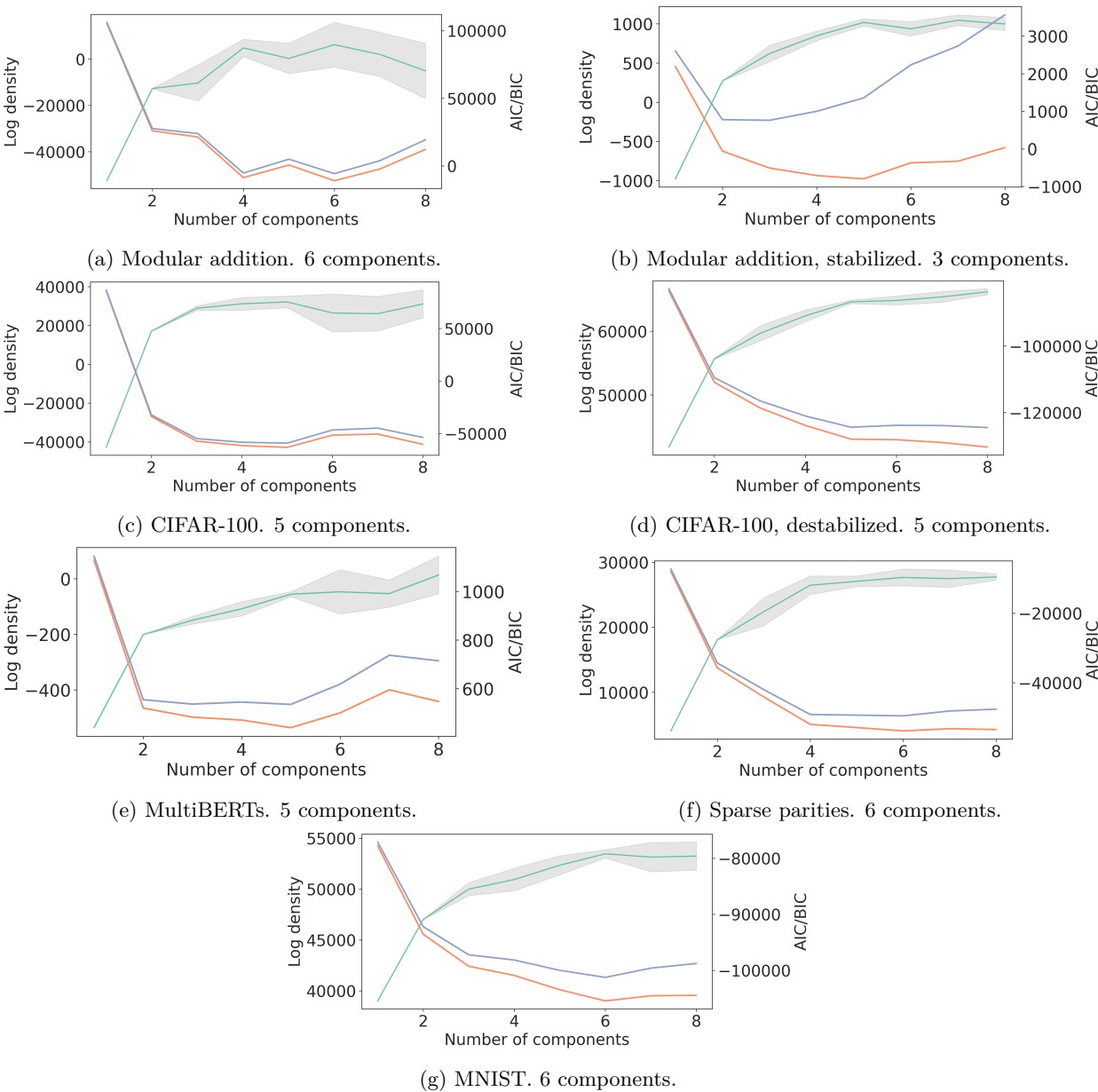

(a) Modular addition. 6 components.

(b) Modular addition, stabilized. 3 components.

(c) CIFAR-100. 5 components.

(d) CIFAR-100, destabilized. 5 components.

(e) MultiBERTs. 5 components.

(f) Sparse parities. 6 components.

(g) MNIST. 6 components.

Figure 10: Model selection curves for choosing the number of hidden states in the HMM. We choose the model with minimum BIC in all cases.

# I  Convergence Time Histograms

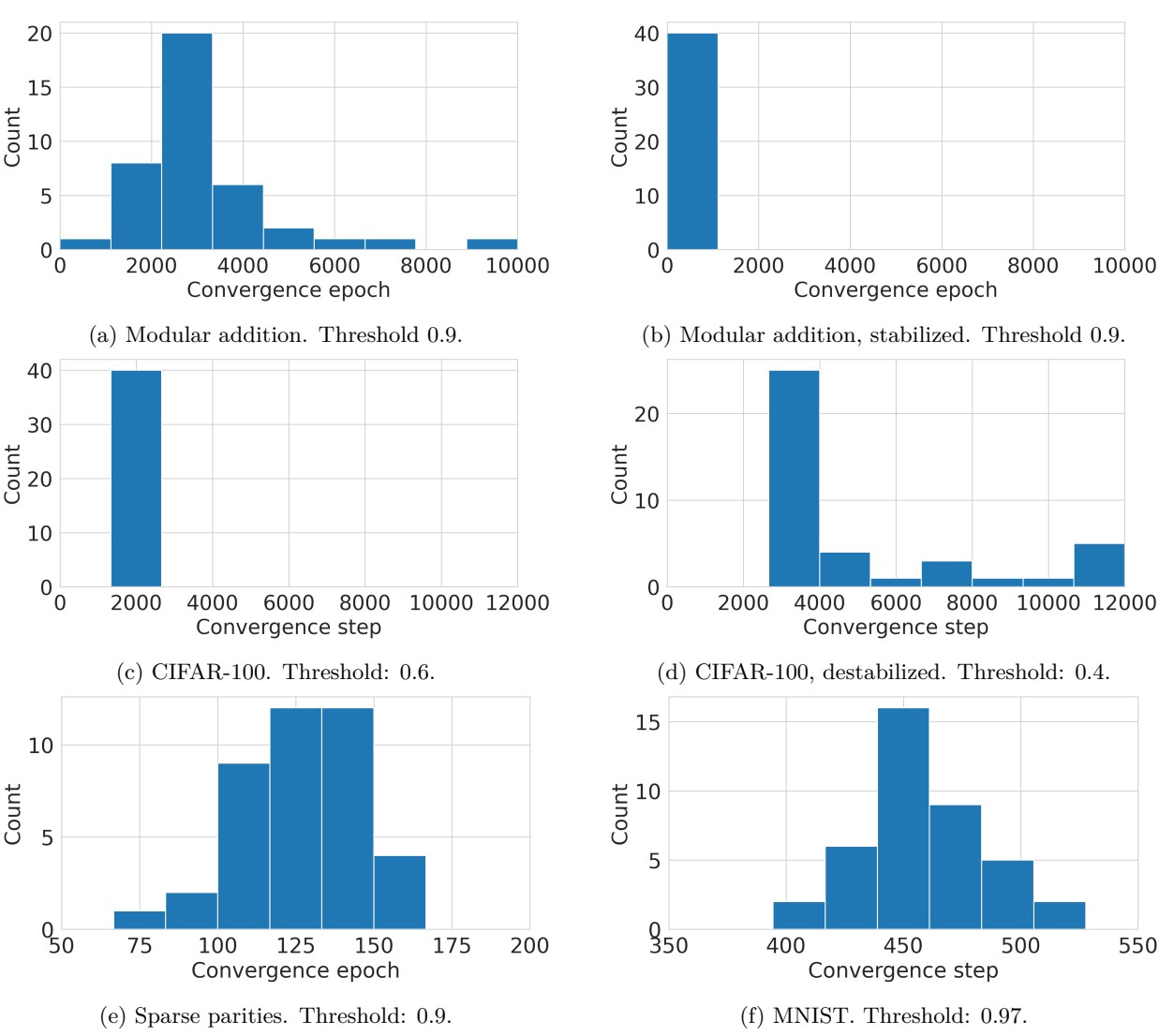

(a) Modular addition. Threshold 0.9.

(b) Modular addition, stabilized. Threshold 0.9.

(c) CIFAR-100. Threshold: 0.6.

(d) CIFAR-100, destabilized. Threshold: 0.4.

(e) Sparse parities. Threshold: 0.9.

(f) MNIST. Threshold: 0.97.

Figure 11: Visualization of convergence times. Convergence time here is defined as the first time a model crosses some threshold of evaluation accuracy, and choose the threshold to be a value slightly less than the performance that the final model achieves. For example, our fully trained models generally achieve perfect accuracy on modular addition, so we choose a threshold of 0.9.

