# OpenReview forum: "Latent State Models of Training Dynamics"
_TMLR — Accepted by TMLR_

### Review · Reviewer_ZHnM · 2023-09-07

**Summary Of Contributions:**

This paper studied the effect of randomness on the training dynamics of neural networks. The author used a sequence of metrics on the training trajectories and established a hidden Markov model (HMM) to analyze the variations from random seeds during different training processes. Specifically, the authors presented that modular arithmetic and sparse parity experiments have grokking phenomena and are affected by randomness, while the image classification experiments with batch normalization and residual connections are insensitive to random seeds. The authors proposed detour states in HMM and showed that detour states appear when the model is sensitive to randomness and otherwise disappears.

**Audience:**

Yes

**Claims And Evidence:**

Yes

**Requested Changes:**

See above.

**Strengths And Weaknesses:**

Strength:
1. Whether the training process of the neural network is robust to randomness or not is an important question to study but so far not too much analysis has been done. This paper proposes an interesting method using HMM to analyze the training dynamics of neural networks. The authors suggested running each model with more training seeds to control the variation in the training processes.

2. This paper provides some new insight into the grokking phenomenon. The experiments showed that adding layer normalization, residual connections, and decreasing batch size may help us smooth the loss landscape and avoid the grokking phenomenon.

Weakness:
1. There is no explanation of the choices of different metrics in Appendix B. Why can these metrics interpret the features of the neural networks during training? For very large models, some of these metrics cannot be efficiently recorded during training.

2. The method proposed in this paper using HMM may be computationally inefficient for large models. This method requires repeatedly training neural networks many times with different random seeds, which becomes expensive for analyzing large models in practice. Any comments for this on training HMM and extracting the training map?

Minor and question:
1. Before section 2.1, $d<<D$ should be $d\ll D$; $f:\mathbb{R}^D\to \mathbb{R}^d$ should be $f_i:\mathbb{R}^D\to \mathbb{R}^d,$ $1\le i\le d$.
2. Notation \{$z_{1:T}$\}$1^N$ on page 3 section 2.1 is confusing. I understand you consider $N$ random seeds and for each seed, you have one $z_{1:T}$.
3. In conclusion, the authors claim that stability to randomness is linked to generalization. Can you explain this more from the experiments in sections 3.-3.33? For instance, in Figure 2, although we observe the grokking phenomenon, what would be the final generalization error for each case? Can we get similar and comparable test errors if we train the neural network sufficiently longer time? Additional explanations may be needed for these experiments.
4. Here are some additional references that may be related to this work. [1,2] also studied the training dynamics of the neural networks using different metrics. [3] also studied the grokking phenomenon in sparse parity tasks. [4,5] are some other references for the effect of randomness in neural networks.

[1] Deep learning versus kernel learning: An empirical study of loss landscape geometry and the time evolution of the Neural Tangent Kernel.
[2] Spectral evolution and invariance in linear-width neural networks.
[3] A Tale of Two Circuits: Grokking as Competition of Sparse and Dense Subnetworks.
[4] Effects of Random Seeds on the Accuracy of Convolutional Neural Networks.
[5] torch.manual seed(3407) is all you need: On the influence of random seeds in deep learning architectures for computer

---

> ### Author Response · Authors · 2023-09-19
> **Response to Reviewer ZHnM**
>
> Dear ZHnM,
>
> Thank you for the helpful comments! Our responses and offered revisions are below.
>
> >There is no explanation of the choices of different metrics in Appendix B. Why can these metrics interpret the features of the neural networks during training? For very large models, some of these metrics cannot be efficiently recorded during training.
>
> We will extend our discussion of metrics in Appendix B. Broadly, the metrics we chose either capture how weights are dispersed in space or properties of the function computed by a layer. For example, we compute the spectral norm (maximum singular value), averaged across layers, which describes the maximum amount that a vector might change as it passes through the layers of the neural network. This statistic tracks how the computation of the neural network changes during training.
>
> Some metrics, such as the ones related to singular values, can indeed become inefficient to compute for very large models. In these cases, we recommend calculating calculating metrics on a lower-dimensional sketch of the weights [Frequent Directions, Ghashami 2015], using random projections, or using streaming algorithms such as streaming SVD. Users can also try our method without the metrics related to singular values.
>
> > The method proposed in this paper using HMM may be computationally inefficient for large models. This method requires repeatedly training neural networks many times with different random seeds, which becomes expensive for analyzing large models in practice. Any comments for this on training HMM and extracting the training map?
>
> The reviewer is correct here; we will extend our discussion in Limitations. In future work, we hope to train a single training dynamics model that can generalize across training setups. For example, we can consider training a dynamics model over a few language model training runs on different random seeds. We can then use this dynamics model to analyze new language model training runs.
>
> In essence, the cost of collecting training runs may only need to be paid once. We anticipate that using a pretrained dynamics model to analyze new training runs will be a fruitful application of our method.
>
> Another opportunity for future work is testing whether we can extract the key information from training maps—the degree of stability and the presence of detour states—by only analyzing the early stages of training runs. An approach like this could scale to large models while still being useful for making quick decisions during hyperparameter sweeps.
>
> > Minor and question:
> 1. Fixed.
> 2. Fixed.
> 3. We have deleted this last sentence. What we meant was that stability across random seeds is related to stability in convergence time. But this is also stated in the Limitations and Future Work section, so we simply removed the redundancy.
> 4. Added to related work.
>
> Thanks again!

---

### Review · Reviewer_5mBE · 2023-09-20

**Summary Of Contributions:**

The authors propose a study on the effects of randomness in neural network training.
To accomplish this, they fit an HMM where observations are vectors of hand-chosen metrics computed from the checkpoints of a neural network during training. The number of hidden states is a hyper-parameter.

The main contribution is the identification of so called "detour states", which, in their words are "latent states associated with slower convergence".

**Audience:**

Yes

**Broader Impact Concerns:**

No concerns.

**Claims And Evidence:**

Yes

**Requested Changes:**

Critical for acceptance.
- The authors should provide baselines for all experiments, or justify why a baseline can not be provided.
- Please make the derivations more obvious (verbose).

**Strengths And Weaknesses:**

# Strengths
- The paper is nicely presented, with well-chosen figures that clearly convey the concepts.
- Experiments are carefully constructed.

# Weaknesses/Questions

My main issue with this paper is there are no baseline methods for any experiments.

Here are some section specific comments:

Section 2.2
- What does it mean for an edge to be "unused by the HMM for all training trajectories"?
- Should the $\arg\max$ be over $i$ instead of $\tilde{z}[i]$? The states in Figure 1 are labelled with integers.
- Intuition is lacking as to why the difference in means represents a reasonable quantity to use as a label for the edges.

Section 2.3
- What is meant by "probabilistic reasoning"?

Section 3
- I can understand the motivation to leave out embedding matrices when computing metrics, but not layer norms. Isn't the technique part of the neural network architecture choice?

Appendix A

Lemma
- I don't understand why
$p(\tilde{z}\_{1:t}) = \sum\_{i=1}^t \alpha\_i(s_i) = \sum\_{i=1}^t p(s_i, \tilde{z}\_{1:i})$. If the summation was to marginalize out $s_i$, then it would make sense. But it's not clear why this is true when summing over $i$ .

Proposition 1
- What happened to $E$ when going from the 2nd to the 3rd equality? I understand $E$ won't affect optimizations that use the log likelihood as an objective, but that isn't the same as saying the 2nd and 3rd line are equal.

---

> ### Author Response · Authors · 2023-10-05
>
> Many thanks for your suggestions! We have added baselines to the revision and additional explanations to our derivations.
>
> > My main issue with this paper is there are no baseline methods for any experiments.
>
> We have added two baselines for all experiments: 1) an HMM trained on just the L2 norm and 2) k-means over the given statistics. Baseline 1 conveys the importance of using multiple statistics, and Baseline 2 does not take into account temporal structure. These baselines are in Appendix C. Both baselines tend to use significantly more latent states to represent the same training dynamics, which we view as detrimental for interpreting the dynamics.
>
> > Section 2.2
>
> > What does it mean for an edge to be "unused by the HMM for all training trajectories"?
>
> We have rephrased this line. The Markov chain is represented as an adjacency matrix in the HMM. If an edge does not appear in any latent state trajectories. then we set the edge’s value in the adjacency matrix to zero.
>
> > Should the argmax be over i instead of z[i]? The states in Figure 1 are labeled with integers.
>
> The argmax here is meant to choose a specific feature value, so we take the argmax over z_t(i) instead of i. The integers in figure 1 correspond to the hidden state $k$, defined in Section 2.2, second paragraph.
>
> > Intuition is lacking as to why the difference in means represents a reasonable quantity to use as a label for the edges.
>
> We have added an additional sentence explaining this in section 2.2.
>
> > Section 2.3
> > What is meant by "probabilistic reasoning"?
>
> This was vague; we have deleted this phrase.
>
> > Section 3
>
> > I can understand the motivation to leave out embedding matrices when computing metrics, but not layer norms. Isn't the technique part of the neural network architecture choice?
>
> Normalization layers indeed are part of the architecture choice, but they typically only have 1 to 2 learnable parameters. So computing values like the L1 or L2 norm here would be inappropriate. We opted to only include the layers of the neural network that are performing matrix operations; embedding matrices and normalization layers perform lookups and scalings, respectively.
>
> As demonstrated in Section 3.3, the absence of normalization layers both affects training dynamics and the resulting training map, so our method is still sensitive to normalization layers despite not explicitly including information about normalization layer parameters.
>
> > Appendix A
> > Lemma
>
> Our mistake, the sum is indeed over $s_i$. We have edited the notation.
>
> > Proposition 1. What happened to the constant E?
>
> You are correct. We added E back in the last equality.
>
> > The authors should provide baselines for all experiments, or justify why a baseline can not be provided.
>
> Please see appendix C.
>
> > Please make the derivations more obvious (verbose).
>
> We have added additional explanations to Appendix A.
>
> Thank you!

---

### Review · Reviewer_JCA7 · 2023-09-27

**Summary Of Contributions:**

The authors describe a series of experiments where Gaussian HMMs are applied to trajectories of neural network training. The trajectories itself are made of features of the weights of neural networks. Subsequently, certain structures emerge, which the authors trace back to different favourable or less favourable training runs and architectural choices. The authors then identify "detour" states, which they deem hurtful for convergence time.
Additionally, they employ various statistical experiments, such as predicting convergence time or the variety of training runs through a dissimilarity measure induced by the initialisation of the networks' weights.

**Audience:**

Yes

**Broader Impact Concerns:**

I don't think the submission requires a broader impact statement.

**Claims And Evidence:**

No

**Requested Changes:**

The choice of HMM needs to be justified further. I think the model is not adequate, as the unobserved dynamics do not follow discrete states. What makes an HMM better than, say, a switching LDS, apart from the authors (and readers!) possibly being more familiar with it? I might even go further and say "detour states don't exist, it's better explained by a flat region". An HMM implies a detour state is a region from which it escapes randomly, while a flat region would mean it needs to work its way through it. Does this have any implications for my day to day work? I don't know, but I wish this paper would tackle such questions.
Yet, since all models are wrong and some are useful, the HMM might still be a reasonable choice. However, this would need support. A section on whether an HMM even is an adequate model–something like, "let's check if this is a good model for the trajectories". For example, I would like to see a comparison of samples from the model and the empirical data?

To take a view point from the other side, let us look at Figure 2, where we have different learning curves. Of course, an HMM captures the differences, but so does a simple graphical illustration. Do we even need an HMM for analysis? Don't we already see all that is necessary by noting the plateaus during training? Isn't the message "some training runs plateau, other's dont–check your hyper parameters and run more experiments to find the best set!" sufficient? Does this approach bring anything to the table? I am not convinced at this stage!

The contributions and findings need to be worked out better. Previewing more of the machinery in the earlier sections would certainly help.
Overall, I have problems finding a proper thread. For example, the trajectory similarity measure and the regression are not mentioned in the preview at the end of the introduction. They take up quite some space though, and somewhat come as a surprise during the first reading.

The figures are very large sometimes and make lots of page turning necessary. Further, the formatting of those–especially the fonts–is not consistent.

Proposition 1 seems like the definition of a derivative to me. I don't think putting this fact into a formal framework helps. The same goes for the definition of the detour state.

Further, I think the evidence needs to be presented better. I must repeat that I felt very confused at times and felt that I am overloaded with unstructured information. I wish I could provide more constructive feedback at this point with concrete measures.

**Strengths And Weaknesses:**

I enjoyed the application of a probabilistic model to the training process. After all, researchers are subject to high-dimensional dynamic systems all the time during the attempts to make a new model work. Still, curricula are very short on the subject and knowledge about it is passed on as lore from the elder's mouth to the disciple's ear. Hence, applying the fields own tools to the fiels own problems seems like a good idea.

The paper contains lots of experimental evidence and offers a systematic statistical approach to most questions that pop up. Nevertheless, I am not convinced by the overall "package": it left me mostly confused. The first sentence of the conclusion sums it up nicely: "We make several main contributions." I must say, the authors claim many different contributions in a rather unstructured and ad hoc way. A lack of focus emerges and that left me wondering what exactly I learned from the text. The feeling prevailed even after I spent considerable time on the manuscript.

Further, I am not convinced that many of the results are unknown. For example, we know that certain architectural tweaks (e.g. batch normalization) improve learning–the papers introducing these techniques generally do a good job at providing evidence. Something that I personally did not consider before was whether the sensitivity to the initialization implies something about generalization. But, to be honest, the evidence for this claim is either rather weak or not presented well/extensively in the manuscript.

---

> ### Author Response · Authors · 2023-10-05
>
> Thank you very much for your review! We have posted a revision, which we hope gets at your concerns. Please find our specific responses below.
>
> > The choice of HMM needs to be justified further.
>
> To our knowledge, the idea of explicitly modeling training dynamics with an interpretable probabilistic model is novel. In establishing this new category, we chose to examine how far we can push a simple model.
>
> The HMM is linear, Markovian, and has a discrete latent space. The linear assumption we accept as a typical first order approximation. The Markov property depends on the training setup–training with SGD is Markovian, but training with Adam is not. Most importantly, we choose a discrete state space because of observations in previous works (Olsson 2022, Nanda 2023), showing that learning seems to exhibit a few discrete, qualitatively distinct states. We have extended this discussion in the methods section.
>
> Furthermore, we show that the HMM, across all settings:
> - recovers information related to the loss, despite not being trained on it. This allows us to:
>   - connect changes in the loss to changes in specific metrics.
>   - see which changes in metrics are actually important, via arguments in Sec 2.2.
> - learns a Markov chain that can be used to predict convergence time.
>
> From this, we conclude that the HMM learns a useful, low-dimensional representation of training dynamics, despite being a simple model.
>
> > Of course, an HMM captures the differences, but so does a simple graphical illustration. Do we even need an HMM for analysis?
>
> We show that the HMM, despite not being trained on any loss information, segments the training trajectory in a way that is consistent with the loss behavior. Thus, the HMM can be used to connect changes in the loss with changes in the metrics over weights. We make these connections in Section 3.1 and 3.2, where we discuss the various paths taken through the training map and the changes in metrics that typify each path.
>
> It may be possible to examine metrics from all training runs by hand, but this is not systematic, invites human error, and requires sifting through a lot of data by hand. The HMM helps us systematically understand why certain random seeds are more successful. Without the HMM, one loses the connections from performance differences to model internals.
>
> We added a comment on this in the discussion, Section 5.2.
>
> > The contributions and findings need to be worked out better. Previewing more of the machinery in the earlier sections would certainly help.
>
> Point taken, we have added additional signposts to the introduction and methods section.
>
> > The figures are very large sometimes and make lots of page turning necessary. Further, the formatting of those–especially the fonts–is not consistent.
>
> We have made the figures slightly smaller and moved them around slightly to keep more text on the same page.
>
> > Proposition 1 seems like the definition of a derivative to me. I don't think putting this fact into a formal framework helps. The same goes for the definition of the detour state.
>
> We agree and have reformatted these definitions in the revision.
>
> > Further, I think the evidence needs to be presented better. I must repeat that I felt very confused at times and felt that I am overloaded with unstructured information.
>
> We have added key takeaways and stronger topic sentences to the results section.
>
> Thank you for the helpful comments!

---

### Author Response · Authors · 2023-10-05
**High-level comments**

We greatly appreciated the reviewers’ constructive comments. We have posted a revision addressing your comments.

At a high level, here are the contributions of our paper:
- We use a probabilistic model (HMM) to analyze the training dynamics of neural networks. Previous works generally analyze training dynamics by hand.
- Using the HMM, we find detour states, which are latent states that are harmful for convergence in training.
- We show that detour states can be avoided by optimizing training hyperparameters or model architecture. In the grokking settings, these findings are novel. We also draw a parallel from grokking tasks to image classification, a task that has been studied and optimized more extensively by the ML community.

Thank you!

---

### Decision · Action_Editor_nkP7 · 2023-11-09

**Recommendation:** Accept as is

**Comment:**

The paper confronts an important and difficult problem in deep learning practice with a new analysis approach using a state space model (HMM) representation of the learning dynamics. Any attempt to progress the understanding of learning dynamics should be very much welcomed so even though this paper is merely a step on the way, it still holds enough insights to warrant publication despite a bit of misgivings from the referee.

The paper is accepted as is. But please remember to remove any highlighting in the final version.

**Audience:**

Yes.

**Claims And Evidence:**

Yes.